# Adherence to IMCI guidelines for key symptoms in Ethiopian children: A 2021–2022 national service provision survey

**Abiyu Abadi Tareke**[1]*, **Awoke Keleb**[2], **Kaleab Mesfin Abera**[3], **Natnael Kebede**[4], **Endalkachew Mesfin**[3], **Aznamariam Ayres**[5], **Yawkal Tsega**[3], **Abel Endawkie**[5], **Shimels Derso Kebede**[6], **Eyob Tilahun Abeje**[5], **Ermias Bekele Enyew**[6], **Chala Daba**[2], **Lakew Asmare**[7], **Fekade Demeke Bayou**[6], **Mastewal Arefaynie**[8], **Asnakew Molla Mekonen**[4]

**1** Amref Health in Africa, COVID-19 Vaccine/EPI Technical Assistant at West Gondar Zonal Health Department, Gondar, Ethiopia, **2** Department of Environmental Health College of Medicine and Health Sciences, Wollo University, Dessie, Ethiopia, **3** Department of Health System and Management, School of Public Health, College of Medicine and Health Sciences, Wollo University, Dessie, Ethiopia, **4** Department of Health Promotion, School of Public Health, College of Medicine and Health Sciences, Wollo University, Dessie, Ethiopia, **5** Department of Epidemiology and Biostatistics School of Public Health College of Medicine and Health Science Wollo University, Dessie, Ethiopia, **6** Department of Health Informatics, School of Public Health, College of Medicine and Health Sciences, Wollo University, Dessie, Ethiopia, **7** Department of Epidemiology and Biostatistics, Institute of Public Health, College of Medicine and Health Sciences, University of Gondar, Gondar, Ethiopia, **8** Department of Reproductive and Family Health, School of Public Health, College of Medicine and Health Sciences, Wollo University, Dessie, Ethiopia

* abiyu20010@gmail.com

**Data Availability Statement:** All data used to conduct this study are provided within the manuscript.

## Abstract

### Background

The Health Services Provision Assessment in Ethiopia (SPA-ET) is a survey that generates data on the availability and quality of health services in Ethiopia. Despite the presence of integrated management of childhood illness guidelines in healthcare settings, there has been inadequate exploration or assessment of how effectively and consistently health professionals follow the guidelines.

### Objective

This study aims to identify factors influencing healthcare worker adherence to the integrated management of childhood illness guidelines to identify spatial clusters.

### Methods

The data for this study were gathered from the Service Provision Assessment (SPA) survey in Ethiopia, which was conducted nationwide from August 11, 2021, to February 4, 2022. It included a total of 788 health professionals who assessed sick children experiencing at least one of the three main childhood illness symptoms: fever, cough, or diarrhea. We employed STATA version 16 for data analysis, utilizing cross-tabulations to explore relationships between variables and logistic regression modeling to identify factors influencing adherence. To account for the hierarchical structure of the health survey data, we employed

**Funding:** The author(s) received no specific funding for this work.

**Competing interests:** The authors have declared that no competing interests exist.

**Abbreviations: AIC**, Akaike information criteria; **aOR**, Adjusted Odds Ratio; **BIC**, Bayesian information criteria; **CI**, confidence interval; **COR**, Crude Odds Ratio; **DIC**, Deviance Information Criterion; **EA**, enumeration areas; **EPHI**, Ethiopian public health institute; **ESPA**, Ethiopian Service Provision Assessment; **HC**, health center; **HF**, health facility; **HO**, health officer; **ICC**, Intra-Class Correlation; **ID/EA**, cluster ID/ enumeration areas; **IMCI**, Integrated Management Childhood Illness; **IQR**, interquartile range; **KM**, Kilometer; **LLR**, Log-Likelihood Ratio; **MD**, Medical Doctor; MOH, Ministry of Health; **NCD**, Non-Communicable Diseases; **NTD**, Neglected Tropical Diseases; **PCV**, Proportion of Variance Change; **SNNP**, South Nation and Nationalities People; TB, tuberculosis; **USAID**, United States Agency for International Development; **VIF**, variance inflation factors.

multilevel logistic regression. Model selection was based on comparison parameters including the Bayesian Information Criterion (BIC) and Akaike Information Criterion (AIC). We computed adjusted odds ratios with 95% confidence intervals, and statistical significance was determined at a significance level of $p < 0.05$.

## Results

The rate of adherence to the integrated management of childhood illness guideline was 33% (95% CI: 29.70%, 36.26%). The analysis revealed several factors influencing adherence to IMCI protocols. child's age (being ≥24 months) [aOR = 0.66, 95% CI: (0.45, 0.87)], facility type (health center) [aOR = 2.61, 95% CI: (1.84, 3.37)], place of residency (being rural) [aOR = 0.54, 95% CI: (0.38, 0.77)], and care provider's qualification (health officer) [aOR = 1.71, 95% CI: (1.18, 2.48)] were all statistically significant. Moreover, the primary cluster is situated in the west Oromia region, with a central focus at coordinates (7.982108 N, 36.203355 E) and extends to a radius of 78.28 km.

## Conclusion

This study confirms a low adherence rate (33%) among health professionals in Ethiopia to the IMCI guideline for assessing the three main symptoms of sick children. The study identified child's age, facility type, academic qualification, and place of residence as crucial factors correlated with adherence rate. Furthermore, 5 secondary clusters (hotspot areas) were identified using SaTScan software. To address the higher protocol refusal, interventional plan needs to be based on academic qualification of care provider, facility type, age of child and place of residency. Moreover, interventions to reduce non-adherence to IMCI guidelines should be location-tailored based on identified hotspot areas to restore guidelines adherence equality.

## Introduction

The Ethiopian health service provision assessment (E-SPA) is a holistic survey mainly conducted to assess the quality and availability of health services at innumerable healthcare facilities (hospitals, health centers and other private health facilities) [1]. The survey provides information regarding the performance of health systems, weaknesses and strengths of the healthcare delivery system, and areas of enhancement. SPA also assesses the quality of healthcare by evaluating the adherence of health workers to clinical guidelines such as integrated management of child illness (IMCI), patient management and infection prevention [2].

To assist health workers in effectively diagnosing and treating sick children in laboratories and medical equipment limited countries like Ethiopia, World Health Organization (WHO) and the United Nations Children's Fund (UNICEF) introduced the IMCI guideline in 1995 [3–6]. IMCI has countless potential to improve healthcare providers performance [7, 8] and quality care for sick children [3]. In Ethiopia, a country with several healthcare challenges [9], IMCI guidelines have been instrumental in guiding health professionals in the assessment and management of these critical symptoms [8].

This guideline starts with identification of key symptoms of sick children. Among these, the guideline mainly assesses three symptoms which are indicative of a range of common

childhood illnesses, including pneumonia, diarrhea, respiratory infections, and other potentially severe conditions. These symptoms can serve as red flags, potentially necessitating hospitalization [10]. Childhood illnesses, particularly those presenting with vital symptoms like fever, diarrhea, and cough/difficulty of breathing, are a significant contributor to the global burden of illness and death among children under five [11].

In 2015, preventable diseases such as pneumonia, diarrhea, and malaria tragically claimed the lives of an estimated 16,000 children under five daily [12]. Current data in Ethiopia indicates high child mortality rates: 29 neonatal deaths, 48 infant deaths, and 67 under-5 deaths per 1,000 live births [13]. Despite the high burden of child mortality and morbidity, surveys conducted in health facilities across three Ethiopian regions—Amhara, Oromia, and Southern Nations, Nationalities, and Peoples' Region (SNNP)–revealed disappointingly low coverage of IMCI guidelines. Amhara exhibited a 20% IMCI coverage, Oromia recorded a mere 4% coverage, and the SNNP Region reported a 25% coverage [14].

While IMCI has been scaled up in all health facilities across Ethiopia, there remains a significant gap in understanding the adherence status of health professionals to this guideline. And there has been limited exploration or assessment of how effectively and consistently health professionals are following these guidelines. Our literature review found no prior research on healthcare worker adherence to IMCI protocols in Ethiopia.

This study is the first to assess healthcare worker adherence to IMCI guidelines for evaluating the three key childhood illness symptoms. To gain a robust understanding of healthcare worker adherence to IMCI guidelines across Ethiopia, this research utilizes a comprehensive analysis of the most recent data. Going beyond national-level assessment, this study explores spatial patterns by employing the SaTScanTM software tool. This powerful tool allows for the identification of hotspots with high concentrations of non-adherent healthcare professionals, enabling targeted interventions to improve adherence. By pinpointing hotspot areas, this study offers valuable insights for both national and local health decision-makers. This information allows targeted interventions to be implemented in specific geographic locations, ultimately improving healthcare worker adherence to IMCI guidelines.

Therefore, this study plays a vital role in unmasking IMCI adherence related problems in Ethiopia. By utilizing national representative data from the latest survey and incorporating advanced statistical techniques, our research provides an accurate assessment of the spatial variation and factors linked to IMCI adherence. The findings serve as a valuable resource for policymakers, aiding them in addressing the specific needs of different regions within the country. Through this study, we aim to contribute significantly to the ongoing efforts in maximizing IMCI guideline implementation coverage in Ethiopia.

## Methods and materials

### Study design and period

Ethiopia, the second-most populous country in Africa, has a diverse administrative structure that ranges from the highest level of regions to the lowest level of Kebele. Ethiopia is bordered by Sudan and South Sudan on the west, Eritrea and Djibouti on the northeast, Somalia on the east and southeast, and Kenya on the south. The distribution of population between rural and urban communities is unequal. The total fertility rate of the country is 4.6 births per woman. The corresponding crude birth rate was 32 per 1,000 in 2016. The average household size is 4.6, and the population is projected to reach 122.3 million by 2030 [2].

Ethiopia's healthcare system has three tiers: primary (health posts, centers, and hospitals), secondary (general hospitals serving about a million people), and tertiary (specialized hospitals for 5 million) [15, 16]. While focused on accessibility through primary care, challenges remain

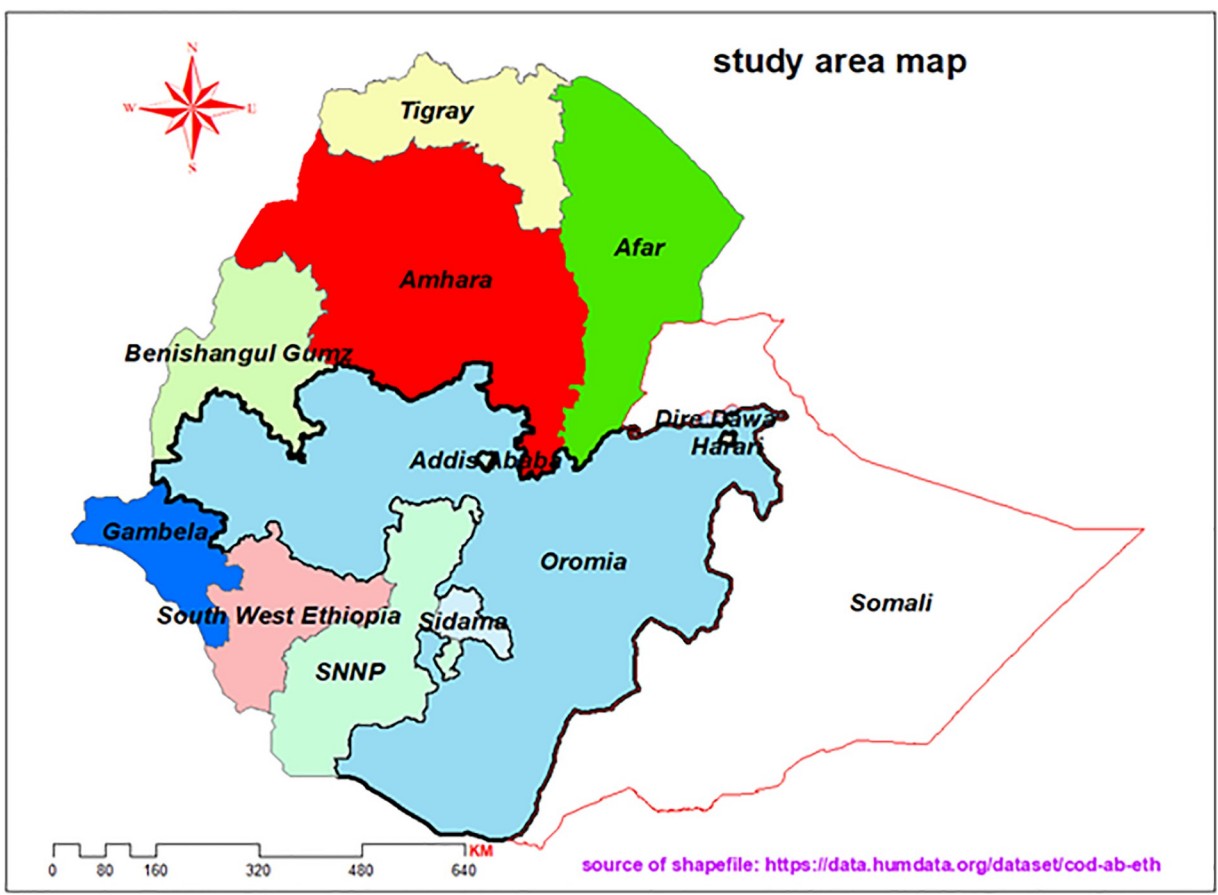

**Fig 1. Map of the study area (Ethiopia) in 2021.**

in service quality and rural access. The healthcare tier system includes both public and private health facilities. Private health facilities include clinics, diagnostic centers, and pharmacies. The country comprises eleven regions, namely Tigray, Oromia, Afar, Sidama, Afar, Amhara, Benishangul Gumuz, south west Ethiopia, SNNP, Gambella, and Somali (Fig 1). Additionally, it encompasses two administrative cities, namely Dire-Dawa and Addis Ababa. Each region is further divided into zones, zone into woredas, and woreda to the last administrative structure known as Kebeles (Fig 1).

## Population

**Source population.**   All healthcare workers in Ethiopia.

**Study population.**   Healthcare workers who were directly involved in the assessment of sick children exhibiting the three main symptoms: fever, diarrhea and cough/difficulty of breathing were included in this study.

**Data sources.**   Data for this study was obtained from the second SPA-ET survey which was collected from August 11, 2021 to February 4, 2022. The survey process was led by the Ethiopian public health institute (EPHI) and ministry of health (MOH) and financed by the United States Agency for International Development (USAID). The main objective of the 2021 SPA-ET was to collect information on the availability and delivery of healthcare services in the country. It also examines the readiness of health facilities to provide quality health services.

The main focus areas of E-SPA were child health, maternal and newborn care, family planning, sexually transmitted infections (STI), HIV/AIDS, tuberculosis (TB), malaria, non-communicable diseases (NCD) and neglected tropical diseases (NTD) [2].

**Study variables.** *Dependent variable.* The dependent variable was adherence to IMCI guidelines, categorized dichotomously as "Yes/ No" variable. The IMCI strategy for diagnosing childhood illnesses emphasizes a systematic approach. Healthcare providers actively inquire about cough/difficulty breathing, diarrhea, and fever, even if caregivers don't mention them, to ensure they don't miss serious conditions. This combines the valuable knowledge of the caregiver with a structured evaluation to effectively assess a child's health.

Adherence was assessed through direct observation of sick child consultations by trained survey interviewers. The interviewers used standardized observation checklists specifically designed to capture adherence to IMCI guidelines for these three key symptoms. But this observational doesn't include physical examination checklists. Checklists included items about history of three main symptoms for any sick child: 1. Did the health professional asked the caregiver if the child have difficulty of breathing or cough? 2. Did the health professional asked the caregiver if the child have diarrhea? and 3. Did the health professional ask the caregiver if the child have fever?

*Adherence.* If healthcare provider is actively asking caregivers specific questions about all three symptoms, regardless of whether they are the main reason for the visit.

*Non-Adherence.* A health professional was classified as non-adherent if he/she failed to ask about at least one of the three symptoms.

## Independent variables

Following a comprehensive literature review, we selected variables to be included in a multi-level multivariable regression analysis. These variables encompass individual-level factors such as healthcare professional's academic qualification, child's age, sex of both the care provider and child, case type (referral/non-referral), and child's place of residence. Additionally, region and facility type were considered as facility-level factors. These variables were identified as crucial factors potentially influencing adherence to IMCI protocols.

**Sample size and sampling procedures.** *Facility sampling procedure.* **Sampling frame.** The sampling frame excluded the Tigray region and consisted of a master list of 25,711 functioning health facilities in Ethiopia, obtained from the Ministry of Health. The facilities included hospitals, health centers, health posts, and private clinics of different designations (higher, medium, lower, and specialty).

*Sampling procedures.* All 413 hospitals (including 41 newly identified ones) were included in the sample. Health centers, clinics, and health posts were sampled using equal probability systematic sampling.

Health centers were slightly oversampled compared to clinics, and clinics were slightly oversampled compared to health posts. Within clinics, higher clinics were all included, medium clinics were slightly oversampled compared to lower clinics, and lower clinics were slightly oversampled compared to other clinics. The final sample included 1,407 health facilities: 413 hospitals, 310 health centers, 328 health posts, and 356 clinics. Due to various reasons, data was successfully collected from 1,158 facilities, representing 82% of the original sample [2].

*Sampling of healthcare providers.* Up to 15 health care providers were interviewed at each facility if available. In total, 21,298 providers were present on the day of the survey, and 8,564 (40%) were selected for the health provider interviews. Of the 8, 564 interviewed and observed healthcare workers,788 (weighted) health professionals assessed sick children during the time of data collection and used for this study.

## Data analysis

*Spatial analyses.* To explore, create, visualize, and edit the spatial information of health workers' IMCI adherence in Ethiopia, we employed Aeronautical Reconnaissance Coverage Geographic Information System (ArcGIS) version 10.8 software. To assess the spatial pattern of health professionals who are non-adherent to IMCI guidelines, we utilized an inferential statistic known as the Spatial Autocorrelation (Global Moran's I) tool [17]. To search for local hotspot areas (local areas with a higher occurrence of non-adherent health professionals to the IMCI guidelines than expected), purely spatial scan statistics were employed using SaTScan™ version 10 software. We retained the default maximum spatial cluster size setting (50% of the at-risk population).

**Multivariable multilevel analysis.** To account for the sharing of similar characteristics at health facilities, we developed a two-level multivariable multilevel logistic regression model in which health workers nested in their facility type (hospital, health center and private health facility). Bivariable analyses were performed to assess the association between IMCI adherence and each exposure variable (one variable at a time) using multivariable multilevel logistic regression analysis. Variables that were considered statistically significant or which displayed an association with IMCI adherence in the bivariable models with a P-value of less than 0.25 were included in the final models (multivariable multilevel regression). Exposure variables were checked for multicollinearity using variance inflation factors (VIF). Variables with VIFs greater than 10 were considered to be indicative of multicollinearity and should be removed from the final models [18].

We developed four models to identify factors associated with IMCI adherence. The fully unconditional model (Model 1) was built with no predictors and aimed to describe the components of the total variance in the odds of IMCI adherence and the intra-class correlation (ICC). The next model (Model 2) was adjusted for individual level (level-one) factors that showed bivariable associations with IMCI adherence. The next model (Model 3) was adjusted for facility-level (level two) factors. The final model (Model 4) was adjusted for both individual and facility-level factors. Adjusted odds ratios with 95% confidence intervals (CI) were used to estimate the association between individual and facility-level factors and the odds of IMCI adherence.

The intra-class correlation was used as a measure of whether the variation in IMCI adherence was primarily between or within facilities. ICC is the ratio of the level-two variance to the total variance and takes a value between 0 and 1, with higher values suggesting greater importance of facility-level factors [19] in understanding the individual differences in IMCI adherence. The proportional change in variance (PCV) was used to measure the change in variance between different models [20]. In this case, PCV was used to measure the total variation in IMCI adherence attributable to individual- and facility-level factors.

Goodness of fit was assessed for each of the four models using Akaike's information criterion (AIC) and Bayesian information criteria (BIC) [21] to select the model with the optimal fit. Lower values of AIC and BIC were indicative of better fit.

**Ethical consideration.** This study used datasets obtained from a nationally representative Service Provision Assessment (SPA) survey. The survey data used in this study were anonymized and could not be traced back to the individual participants. The original data collection process ensured adherence to ethical standards, and appropriate consent was obtained from the participants. Therefore, additional consent was not required for this study.

## Results

From the total of 788 health professionals who assessed sick children, 300 (38%) were from the Oromia region, followed by SNNP 212 (27%) (Table 1). The median age of the children who participated in this study was 18 months (IQR: 9–31 months).

**Table 1. Socio-demographic characteristics of respondents, SPA-ET 2022 (weighted sample, n = 788).**

| Variable | Frequency (weighted sample) | Percent (%) |
|---|---|---|
| **Children's age** | | |
| 2–24 months | 471 | 59.84 |
| ≥24 months | 317 | 40.16 |
| **Sex of child** | | |
| Male | 394 | 49.91 |
| Female | 394 | 50.09 |
| **Children's place of residency** | | |
| Urban | 366 | 46.42 |
| Rural | 422 | 53.58 |
| **Case type** | | |
| Cold case (not referral) | 129 | 16.41 |
| Referral | 659 | 83.59 |
| **Sex of healthcare provider** | | |
| Male | 509 | 64.51 |
| Female | 279 | 35.49 |
| **Qualification of care provider** | | |
| Nurse | 464 | 58.88 |
| HO | 190 | 24.04 |
| MD | 134 | 17.09 |
| **Facility type** | | |
| Private HF | 317 | 40.14 |
| HC | 368 | 46.76 |
| Hospitals | 103 | 13.10 |
| **Region** | | |
| Afar | 3 | 0.31 |
| Amhara | 99 | 12.58 |
| Oromia | 300 | 37.99 |
| Somali | 22 | 2.83 |
| Benishangul | 8 | 1.03 |
| S.N.N. P | 212 | 26.94 |
| Gambella | 8 | 1.05 |
| Harari | 3 | 0.37 |
| Addis Ababa | 70 | 8.85 |
| Dire Dawa | 18 | 2.30 |
| Sidama | 45 | 5.75 |

**NOTE**: HF-health facility, HC- health center, MD- medical doctor, HO- health officer, S.N.N.P- south Nations, Nationalities, and People

The reported adherence rate to IMCI guidelines in Ethiopia was 33% [95% CI: 29.7%, 36.3%]. Notably, certain regions exhibited rates significantly above the national average, reflecting varying percentages of adherence to the IMCI guidelines. Benishangul Gumuz, Dire-Dawa, Harari, and Gambella stand out with increased rates. Oromia (30%) and Sidama (29%) were among the lowest performing regions (Fig 2).

The adherence rate of health workers to the IMCI guideline was highest during the assessment of fever, specifically at 70%. Adherence is higher, specifically 36%, when assessing children aged 2–24 months compared to those aged ≥ 24 months. Furthermore, a highest

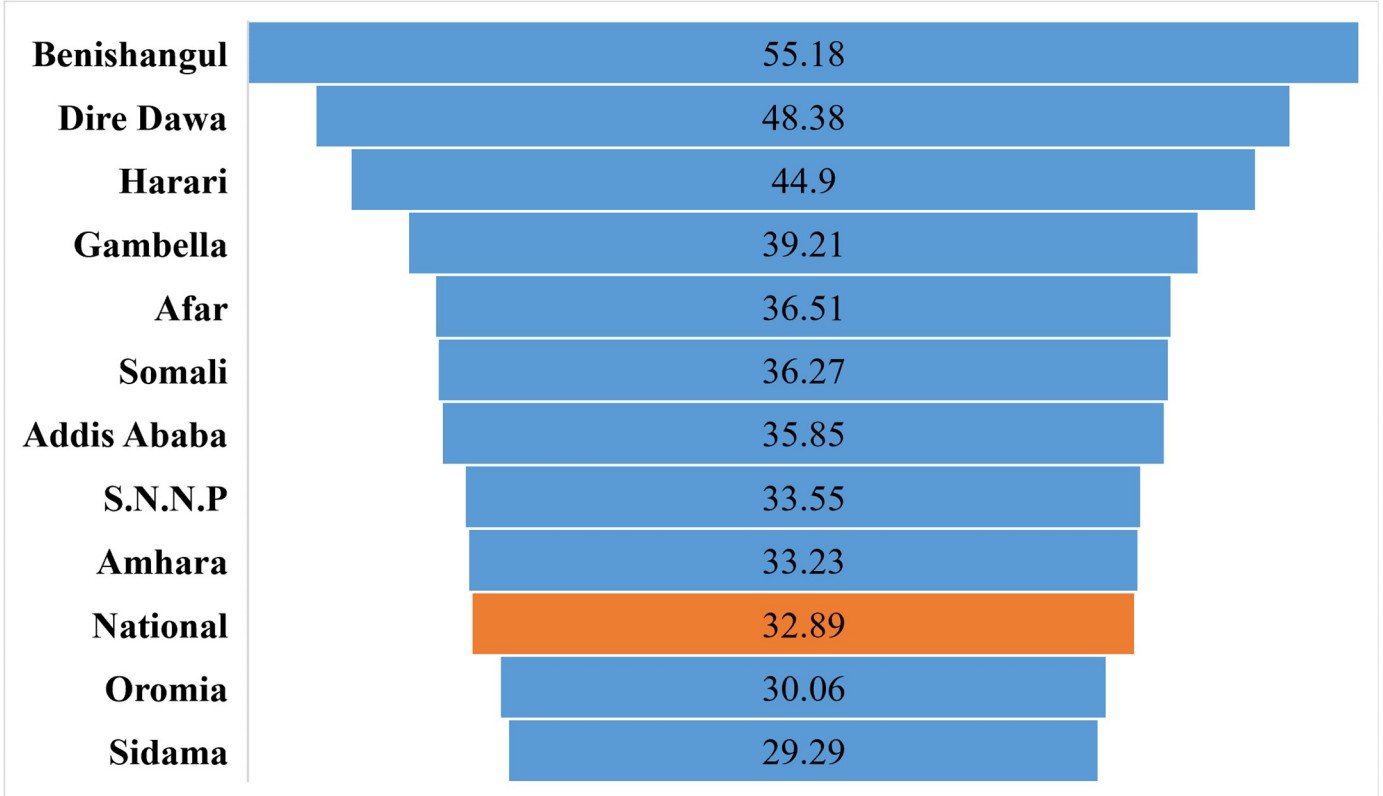

**Fig 2. Health professionals' adherence rate to the IMCI guideline in Ethiopia by administrative region, SPA-ET 2022.**

percentage of adherence, specifically 44%, was observed when children were assessed by health officers. Similarly, children seen at health centers showed an adherence rate of 42%. (Table 2).

## Locations of non-adherent health workers

The SaTScan statistics detected clusters categorized into primary and four secondary clusters. A total of 87 statistically significant location IDs (EAs) were found in those categories of clusters. Out of these, 13 EAs were located inside the window of the most likely cluster (a crucial focus point for targeted interventions) and the remaining clusters were classified as secondary clusters. (Table 3).

**Table 2. Adherence rate of health professionals to the IMCI guidelines while assessing the three symptoms of sick child in Ethiopia, SPA-ET 2022.**

| IMCI items | Overall adherence | Age of child | | Qualification of the provider | | | Facility types | | |
|---|---|---|---|---|---|---|---|---|---|
| | | 2–24 months | ≥24 months | Nurse | HO | MD | Private HFs | HCs | Hospitals |
| Fever | 70.55% | 73% | 64% | 63% | 83% | 80% | 65% | 73% | 77% |
| Diarrhea | 50.38% | 49% | 52% | 49% | 51% | 53% | 45% | 54% | 49% |
| Cough or difficulty of breathing | 60.17% | 61% | 59% | 57% | 67% | 65% | 57% | 61% | 64% |
| All 3 symptoms included | 32.89% | 36% | 28% | 27% | 44% | 36% | 22% | 42% | 33% |

**Note:** HO- health officer, MD-medical doctor, HFs -health facilities, HC-health centers

**Table 3. All clusters of non-adherent health professionals identified through SaTScan in Ethiopia, SPA-ET 2022.**

| Types of clusters | Coordinates/Radius | No. cluster | Cluster location | Expected cases | Observed cases | p-value |
|---|---|---|---|---|---|---|
| Most likely cluster | (7.982108 N, 36.203355 E) / 78.86 km | 13 | West Oromia | 34 | 50 | 0.0000 |
| Secondary cluster 1 | (4.076593 N, 38.317550 E) / 295.43 km | 44 | South Oromia, south-east of SNNP regions | 57 | 80 | 0.0000 |
| Secondary cluster 2 | (10.593845 N, 38.753502 E) / 108.98 km | 21 | Southern Amhara region | 21 | 31 | 0.0006 |
| Secondary cluster 3 | (8.839652 N, 38.693485 E) / 12.45 km | 4 | Addis Ababa | 20 | 30 | 0.0008 |
| Secondary cluster 4 | (7.614975 N, 39.500907 E) / 52.73 km | 5 | Oromia region | 18 | 27 | 0.0027 |

**NOTE**: KM-kilometer, E-east, N-north

SaTScan results indicated that the primary cluster (a cluster with high LLR, 20.82) mainly converged on the west Oromia region (Fig 3), centered at (7.982108 N, 36.203355 E) within a radius of 78.86 km (Table 3). The relative risk of 1.55 can be interpreted as the probability of non-adherence in the primary cluster is 55% higher compared to care provider outside the cluster.

**Multilevel multivariable logistic regression.** *Measures of variation.* An ICC value of 0.5523 in model 1 indicates that more than half (55.23%) of the variability in the odds of IMCI adherence is attributed to differences between facilities. When individual-level factors were

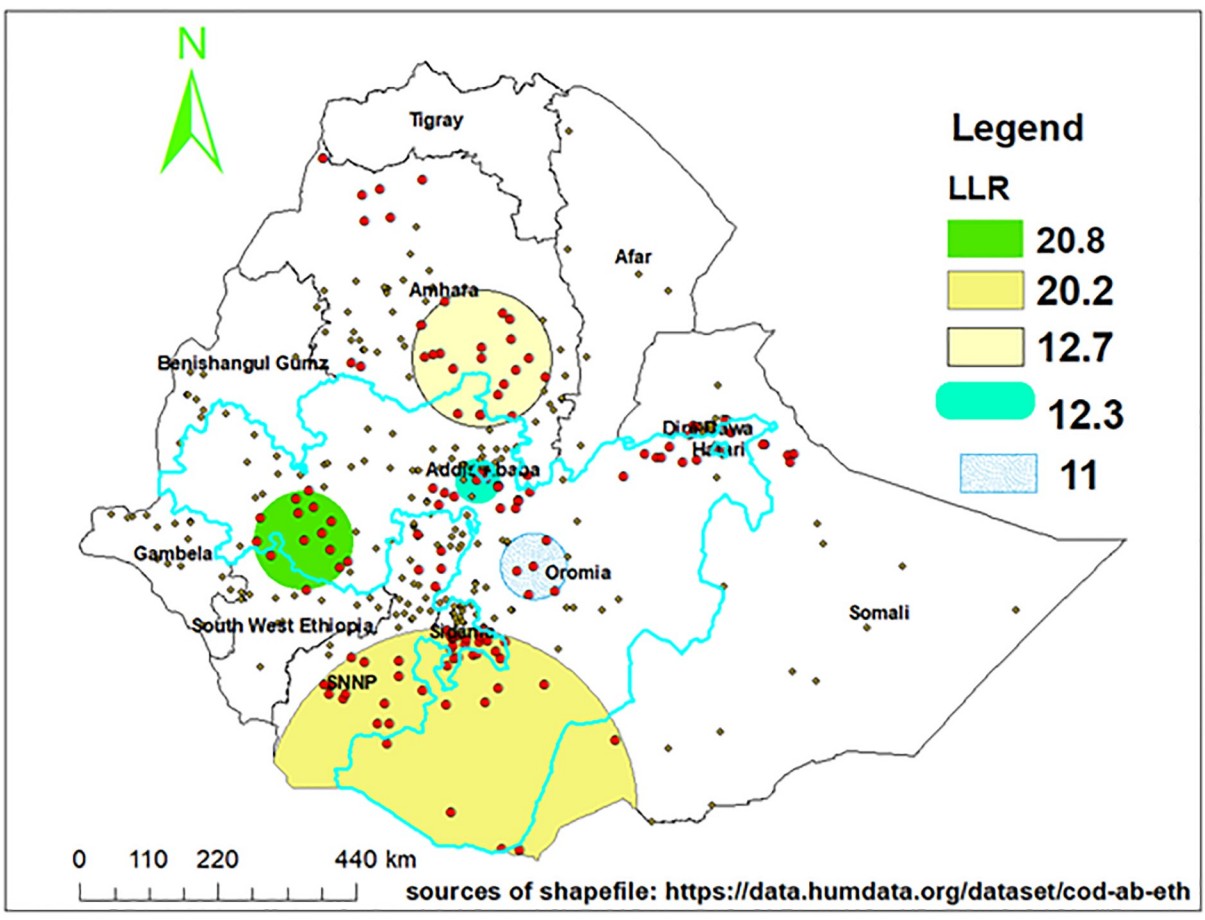

**Fig 3. Location of statistically significant spatial clustering of healthcare providers which are not adherent to IMCI in Ethiopia, SPA-ET 2022.**

**Table 4. Measure of facility level variation in model 1 to 4.**

|  | Model 1 | Model 2 | Model 3 | Model 4 |
|---|---|---|---|---|
| **Measure of variation for IMCI adherence** |  |  |  |  |
| Facility level variance (SE) | 4 (0.86) | 4.17 (0.9) | 3.79 (0.82) | 3.84 (0.85) |
| ICC (%) | 55.23 | 52.11 | 53.58 | 53.89 |
| Explained variance PCV (%) | Ref | 4.25 | 5.25 | 4 |
| **Model fitness** |  |  |  |  |
| AIC | 807 | 795 | 815 | 791 |
| BIC | 816 | 838 | 882 | 844 |

**NOTE**: SE-standard Error, ICC-intra-class correlation coefficient, PCV-proportional change in variance, AIC-Akaike information criterion, BIC-Bayesian information criterion

exclusively considered in the model (Model 2), the corresponding ICC decreased to 52.1%, indicating a reduced contribution of between-individual variation to the explained variance (ICC = 0.521). Model 4, which accounted for both individual-level and facility-level factors, demonstrated a nearly identical proportion of total variance in the odds of IMCI explained by between-facility variation compared to Model 3 (ICC = 0.536). Moreover, the PCV in Model 2 indicated that the addition of the predictors explained an increased proportion of variation in the odds of IMCI adherence (PCV = 4.25).

As illustrated in Table 4, the process of selecting the model with the optimal fit was guided by lower values of AIC and BIC, both of which are indicators of model fit. Accordingly, Model 4, characterized by lower AIC and BIC values, emerged as the best-fitted model for this dataset.

## Factors associated with adherence

In the realm of multivariable modelling, certain factors emerge as noteworthy contributors to the adherence rate of health workers to the IMCI guideline in Ethiopia. It is worth noting that the age of the child (being ≥24 months), the facility type (specifically, being health professional working in health center), the child's place of residency (in rural areas), and the academic qualification of the care provider all demonstrate themselves as significant variables associated with adherence to the IMCI guidelines.

Regarding relationship between age of children and IMCI adherence, children in the higher age group (≥24 months) show a 34% lower likelihood of being assessed based on IMCI guidelines [aOR = 0.66, 95% CI: 0.45, 0.87] compared to those aged less than 24 months. Furthermore, health workers in rural areas were 46% less likely to adhere to the IMCI guidelines compared to those in urban areas [AOR = 0.54, 95% CI: 0.38, 0.77]. This suggests that health workers are less likely to fully adhere to IMCI guidelines when assessing and treating children in rural settings versus urban settings. Regarding academic qualification, a clear difference emerged. Health officers were 1.7 times more likely to follow IMCI guidelines than nurses [AOR = 1.71, 95% CI: 1.18, 2.48]. Analysis of facility-level factors reveals a significantly higher likelihood of adherence to the IMCI guideline. Specifically, healthcare providers in health centers demonstrate a 2.61 times greater adherence to IMCI guidelines [95% CI: 1.84, 3.37] compared to their counterparts in private health facilities (Table 5).

## Discussion

We explore findings from a detailed analysis, including spatial SaTScan, multilevel multivariable analysis, and an examination of IMCI adherence percentages using the national survey

**Table 5. The multivariable logistic analysis (model 4) of adherence rate to IMCI protocol among health professionals in Ethiopia, SPA-ET 2022.**

| Variable | COR (95% CI) | aOR (95% CI) | p-value of aOR |
|---|---|---|---|
| **Child's age in months** | | | |
| ≤24 months(ref) | | | |
| >24 months | 0.67 (0.49, 0.91) | **0.66 (0.45, 0.87)** | **0.004** |
| **facility type** | | | |
| Private HF (ref) | | | |
| Health center | 2.49 (1.77, 3.48) | **2.61 (1.84, 3.37)** | **<0.001** |
| Hospital | 1.72 (1.06, 2.81) | 1.35 (0.70, 2.61) | 0.405 |
| **Case type** | | | |
| None referral (ref) | | | |
| Referral | 0.85 (0.57, 1.29) | 0.91 (0.58, 1.43) | 0.78 |
| **Child's Place of residency** | | | |
| Urban (ref) | | | |
| Rural | 0.55 (0.41, 0.75) | **0.54 (0.38, 0.77)** | **<0.001** |
| **Qualification of care provider** | | | |
| Nurse (ref) | | | |
| Health officer | 2.15 (1.51, 3.06) | **1.71 (1.18, 2.48)** | **0.012** |
| Medical doctor | 1.54 (1.03, 2.32) | 1.26 (0.67, 2.37) | 0.477 |
| **Sex of provider** | | | |
| Male (ref) | | | |
| Female | 0.76 (0.55, 1.04) | 0.89 (0.63, 1.27) | 0.545 |
| **Sex of child** | | | |
| Male (ref) | | | |
| Female | 0.79 (0.58, 1.06) | 0.75 (0.54, 1.03) | 0.076 |
| **Region** | | | |
| Afar (ref) | | | |
| Amhara | 0.46 (0.009, 22.67) | 0.47 (0.008, 25) | 0.712 |
| Oromia | 0.70 (0.015, 32.12) | 0.89 (0.017, 45.84) | 0.957 |
| Somali | 1.12 (0.018, 67.00) | 1.07 (0.016, 71.7) | 0.974 |
| Benishangul | 2.84 (0.033, 243.8) | 9.31 (0.094, 919.6) | 0.341 |
| S.N.N.P | 1.04 (0.022, 49.15) | 1.14 (0.021, 60.88) | 0.946 |
| Gambella | 1.63 (0.02, 123.78) | 2.19 (0.025, 187.3) | 0.729 |
| Harari | 1.71 (0.011, 263.3) | 1.81 (0.010, 319.9) | 0.821 |
| Addis Ababa | 0.89 (0.018, 44.34) | 0.55 (0.099, 31.13) | 0.775 |
| Dire Dawa | 2.28 (0.039, 131.9) | 2.59 (0.04, 168.33) | 0.654 |
| Sidama | 0.84 (0.016, 42.96) | 1.18 (0.02, 68.41) | 0.934 |

NOTE: ref- reference, COR-Crude odds ratio, aOR- adjusted odds ratio, HF-health facility, S.N.N.P- south Nations, Nationalities, and People

data ESPA. These approaches offer a thorough insight into geographic patterns, individual and facility-level determinants, and the overall percentage of IMCI adherence in assessing three childhood illnesses.

According to this study, the adherence of health workers to the IMCI guideline in Ethiopia is low at 32.89% [95% CI: 29.7%, 36.3%]. This prevalence is consistent with research conducted in Tanzania [22] and Kenya [23], which reported a prevalence of 28.4% and 34%, but lower than studies done using the SPA data of Nigeria (48%), Tanzania (50%), Uganda (57%) (19%)

[23]. The variations in IMCI adherence rates may be attributed to differences in socio-demographics, geography, sample sizes, healthcare infrastructure, training programs, and health system disparities between countries. Other contributing factors include the diverse cadres of health workers analyzed in these studies. Additionally, differences may arise from variations in the assessment domains; our study assessed adherence using only the three main symptoms, whereas other studies incorporated danger signs and physical examinations, either separately or in combination with the three main symptoms (22) of childhood illness.

This study identified a significant proportion of healthcare providers showing reluctance to adhere to IMCI guidelines, particularly in geographic areas with a higher number of non-compliant health workers, notably in the Amhara, Afar, SNNP, Sidama, and Oromia regions. This coincides with previous research in Ethiopia which reported inconsistency in the proportion of children correctly assessed according to IMCI guidelines [24]. The concentration of non-adherent health professionals in those areas could stem from insufficient training [25], understaffing, high workload, lack of awareness, and inadequate supervision and monitoring by higher officials. We urge future researchers to conduct a thorough and detailed assessment to identify the potential root causes, to identify challenges and specific barriers contributing to non-adherence in those identified hotspots geographic areas. We recommend policy makers and program implementers to integrate evidence-based policy recommendations to address systemic issues contributing to non-adherence.

In this study, there is an observed inverse relationship between the age of sick children and adherence to IMCI guidelines. This finding is consistent with study done using four African countries' SPA datasets [23]. Research conducted in Tanzania also suggests a possible bias towards younger children [26]. The study found that healthcare providers dedicated more assessment time to children under 5 years old compared to older children. This might be because healthcare providers spend more time assessing younger children than older ones [26]. Another reason might be due to work overload [27, 28] or that healthcare providers are less likely to stick to IMCI recommendations for older children, possibly due to assumptions that older children are more resilient to illness or perceived conditions as less severe. Not sticking to the recommendations of the IMCI guideline during assessment of three main sickness in older children ($\geq$24 months) could lead to suboptimal health outcomes like misclassification of diseases, increased costs, additional complications, and prolonged hospital stays. The health governing body in the country should focus on the need for enhanced training and capacity building for healthcare professionals to ensure that they are equipped with the knowledge and skills required to appropriately manage childhood illnesses in older children.

Another finding of this study is that children living in rural areas of Ethiopia face challenges when it comes to adhering to IMCI guidelines. Rural heightened challenges when it comes to adhering to the IMCI guidelines, particularly in comparison to their urban counterparts. Similar finding is also reported by another study done in the united states of America [29] which concluded that rural dwellers are less likely to get preventive health services based on recommended healthcare guidelines.

The observed disparity in childhood illness management between rural and urban areas could be attributed to several factors. These factors may include a shortage of qualified healthcare professionals in rural settings The observed disparity in childhood illness management between rural and urban areas could be attributed to several factors. These factors may include a shortage of qualified healthcare professionals in rural settings [30, 31], limited access to healthcare facilities, or potentially lower healthcare utilization rates among the rural population compared to their urban counterparts, limited access to healthcare facilities, or potentially lower healthcare utilization rates among the rural population compared to their urban

counterparts [13, 32]. These professionals are concentrated in urban areas compared to rural settings [33].

Addressing education, community awareness, closing the gap of skilled health professional distribution in rural areas related challenges could contribute to narrowing the gap in adherence to IMCI guidelines between rural and urban populations, ultimately improving the health outcomes of children in rural areas.

Our analysis of care provider factors indicates that health officers (HOs) are more likely to adhere to IMCI guidelines recommendations when assessing children compared to nurses. One possible reason for this could be that health officer professionals receive more specialized education on IMCI guidelines during their professional training at university compared to nurse professionals. According to the Ethiopian curriculum, health officers have a wider scope of practices related to clinical skills, including the comprehensive assessment and management of childhood illnesses [34].

Finally, the facility level factors revealed a significant difference in the likelihood of children receiving clinical services in accordance with IMCI recommendations based on the type of facility visited. Children who sought care at health centers were 2.61 times more likely to have all the three main symptoms assessed in accordance with IMCI guidelines, compared to children attended to at private health facilities. This finding is consistent with other studies done in other African countries such in Uganda [35], north-west Ethiopia [36], Nigeria [37, 38], western Kenya [39] which reported that the provision of health services in accordance with their national guidelines is unsatisfactory. It seems possible that these results are due to the major gap of the country in regulating the health workers in private health facilities [40]. Other alternative explanations for this result might be that private health facilities may face resource constraints, limiting their ability to invest in staff training, and other resources required for implementation of IMCI recommendations. This finding suggests that targeted intervention specifically to private health facilities is required to ensure that children, regardless of the type of facility, receive clinical services in accordance with IMCI recommendations.

## Strengths and limitations of the study

Finally, several strengths and limitations can be pinpointed from this study. The main strength of this study is the utilization of most recent national data to study adherence of health workers to IMCI guideline. The authors of this explored the clustering of health professionals reluctant to adhere to IMCI guideline using the emerging statistical analysis called spatial analysis in the context of health systems. Location specific response to health problems based on identified clusters could help in narrowing health inequalities.

The national survey data utilized in this study omits data of Tigray region. So, the exclusion of Tigray region's data from this study might have limited the generalizability of findings to the entire population of the country. The opportunity to gain insight into the Tigray region about prevalence, contributing factors and hotspot areas of adherence to IMCI protocols are missed and this could impair region-specific policy recommendations. In relation to the cross-sectional aspect, SPA Ethiopia encounters analogous constraints to other cross-sectional surveys i.e., cannot be used to determine the cause of something. Hence, we used data from secondary survey, missing other behavioral and cultural factors in this study could compromise model performance. Finally, this study didn't directly measure healthcare professionals' adherence to established protocols for assessing danger signs and conducting physical examinations in children. While the data hints at areas for improvement, it can't decisively determine overall adherence rates or reasons for deviations from protocols.

This study's observational design introduces limitations. The Hawthorne effect, where healthcare providers modify their behavior when aware of observation, could lead to an over-estimation adherence rate to IMCI protocols. Inaccurate or incomplete recall by caregivers might distort the data, hindering a precise assessment of actual adherence practices.

## Conclusion

The rate of adherence to IMCI guideline in Ethiopia is low. Low adherence could have various implications, including potential gaps in the quality of child healthcare and the prevention and management of childhood illnesses. By tailoring interventions to the unique needs of each geographic area, this targeted plan aims to create a more equitable distribution of health professionals adhering to IMCI guidelines, ultimately improving child healthcare outcomes across the region. A full implementation of IMCI programs should consider several key factors to ensure comprehensive and effective coverage like the diverse needs and contexts of children, healthcare facilities, and providers.

Additional research is necessary to conduct thorough analyses in hotspot areas and to examine changes in IMCI adherence over time, exploring spatio-temporal relationships. It is advisable to undertake studies comparing hotspot areas with non-hotspot regions to discern any noteworthy differences or similarities influencing variations in adherence. Furthermore, we suggest that future researchers employ qualitative research methods to gather local perspectives on IMCI adherence in hotspot areas, providing nuanced insights into community beliefs and attitudes.

## Author Contributions

**Conceptualization:** Abiyu Abadi Tareke, Natnael Kebede, Endalkachew Mesfin, Yawkal Tsega, Abel Endawkie, Chala Daba, Lakew Asmare, Fekade Demeke Bayou.

**Data curation:** Abiyu Abadi Tareke.

**Formal analysis:** Abiyu Abadi Tareke, Shimels Derso Kebede.

**Investigation:** Abiyu Abadi Tareke, Mastewal Arefaynie.

**Methodology:** Abiyu Abadi Tareke, Awoke Keleb, Kaleab Mesfin Abera, Aznamariam Ayres.

**Resources:** Eyob Tilahun Abeje.

**Software:** Abiyu Abadi Tareke.

**Supervision:** Abiyu Abadi Tareke, Awoke Keleb, Abel Endawkie, Shimels Derso Kebede, Eyob Tilahun Abeje, Fekade Demeke Bayou.

**Validation:** Abiyu Abadi Tareke, Lakew Asmare, Asnakew Molla Mekonen.

**Visualization:** Abiyu Abadi Tareke, Kaleab Mesfin Abera, Endalkachew Mesfin, Aznamariam Ayres, Yawkal Tsega, Ermias Bekele Enyew, Mastewal Arefaynie.

**Writing – original draft:** Abiyu Abadi Tareke, Awoke Keleb, Kaleab Mesfin Abera, Natnael Kebede, Endalkachew Mesfin, Aznamariam Ayres, Yawkal Tsega, Abel Endawkie, Shimels Derso Kebede, Eyob Tilahun Abeje, Ermias Bekele Enyew, Chala Daba, Lakew Asmare, Fekade Demeke Bayou, Mastewal Arefaynie, Asnakew Molla Mekonen.

**Writing – review & editing:** Abiyu Abadi Tareke, Awoke Keleb, Kaleab Mesfin Abera, Natnael Kebede, Endalkachew Mesfin, Aznamariam Ayres, Yawkal Tsega, Abel Endawkie, Shimels Derso Kebede, Eyob Tilahun Abeje, Ermias Bekele Enyew, Chala Daba, Lakew Asmare, Fekade Demeke Bayou, Mastewal Arefaynie, Asnakew Molla Mekonen.

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
