## [Decision Letter · Decision Letter 0]

10 May 2024

PONE-D-24-03866Adherence of Health Professionals to National IMCI Protocols: A Study on Assessing Three Key Symptoms in Sick Children in Ethiopia - Insights from the 2021 National Service Provision Survey

PLOS ONE

Dear Abiyu Abadi Tareke

Thank you for submitting your manuscript to PLOS ONE. After careful consideration, we feel that it has merit but does not fully meet PLOS ONE’s publication criteria as it currently stands. Therefore, we invite you to submit a revised version of the manuscript that addresses the points raised during the review process.

Dear Author

Kindly address the concerns of Reviewers thoroughly and incorporate all suggested changes carefully so that your work could meet  PLOS ONE criteria for publication.

Please submit your revised manuscript by **June 1st, 2024.** If you will need more time than this to complete your revisions, please reply to this message or contact the journal office at plosone@plos.org. Please include the following items when submitting your revised manuscript:A rebuttal letter that responds to each point raised by the academic editor and reviewer(s). You should upload this letter as a separate file labeled 'Response to Reviewers'.A marked-up copy of your manuscript that highlights changes made to the original version. You should upload this as a separate file labeled 'Revised Manuscript with Track Changes'.An unmarked version of your revised paper without tracked changes. You should upload this as a separate file labeled 'Manuscript'.

We look forward to receiving your revised manuscript.

Kind regards,

Saidul Abrar, MBBS, MPH

Academic Editor

PLOS ONE

Journal Requirements:

Additional Editor Comments (if provided):

Nil

Reviewers' comments:

Reviewer's Responses to Questions

**Comments to the Author**

1. Is the manuscript technically sound, and do the data support the conclusions?

Reviewer #1: Partly

Reviewer #2: Yes

2. Has the statistical analysis been performed appropriately and rigorously? 

Reviewer #1: Yes

Reviewer #2: Yes

3. Have the authors made all data underlying the findings in their manuscript fully available?

Reviewer #1: Yes

Reviewer #2: No

4. Is the manuscript presented in an intelligible fashion and written in standard English?

Reviewer #1: No, the written report lacks coherence and there are a lot of errors in the manuscript. Some of the percentages didn’t add to the total. I have highlighted areas in the manuscript where the sentence need revision.

Reviewer #2: No

5. Review Comments to the Author

Reviewer #1: Kindly provide details of the outcome variable how was this assessed and how the decision was made regarding the non-adherent workers. I've attached a marked PDF with comments for your reference.

Reviewer #2: Dear author(s), thank you for the opportunity to review your article titled “Adherence of Health Professionals to National IMCI Protocols: A Study on Assessing Three Key Symptoms in Sick Children in Ethiopia - Insights from the 2021 National Service Provision Survey”. Overall, I find your article very interesting and relevant for the journal. However, there are certain points which you can work on to further improve your manuscript:

1. Grammar/language: during review, I noticed many typos and certain sentences that would need to be rephrased. In line 142, it is “Health care workers” while in line 144, it is “Healthcare workers”. Line 161, “caretaker” replace it with "caregiver". Line 247, “Therefore, additional consent is not be necessary for this study”. Line 270, “…toward IMCI guideline is found is higher during…”. Line 367, “… difficulty of breathing in older children is (≥24 months) could result…”. Line 378, “In general, rural healthcare settings face a shortage of skilled health professionals in general(26, 27)…” There is repetition of ‘in general’. All of these sentences need to be rephrased. Please pay attention to upper and lower case letters to use them appropriately, particularly in the titles of the tables and figures. Additionally, a thorough review from a language expert is recommended for further corrections.

2. Abstract: “BIC and AIC”? always write a full form of non-standard abbreviations. “…survey conducted in 2021.”, write down the duration of the survey as you did later in the Methods and Materials section.

3. Introduction: In abbreviation to E-SPA, you've written “Ethiopian health service provision”. Assessment word is missed. Para 6, “This study is the first study…”, rephrase this sentence.

4. Methods: Line 125, provide a reference for the health service-delivery structure for the readers to understand the context of it. Define your health facility types here which is used a lot in your research. Considering the secondary nature of your data, mention whether ethics and consent were taken into consideration in the original survey. Follow strobe or other relevant guidelines to arrange your article.

5. Results: Start with the whole survey demographics then the population of interest or only the later one I.e. for 788 patients. Here you have just included the overall participants' demographics of the original survey. Follow one format for writing confidence intervals. Here you write it as “…with a 95% confidence interval ranging from 29.7% to 36.3%” while in another place you mention it as “(95% CI: 29.70%, 36.26%)”. Line 271, age is written as “≤ 24 months” while in tables it is 2-24 months. This changes the meaning of it. Only follow 2-24 months. Line 270-273, rewrite this whole para in correct order. Additionally here, when a no. is greater among all it's called “highest” not higher. Line 305, “… AIC (Akaike Information Criterion)…” is written as Akaike Information Criterion (AIC). Line 313, “…child being over 24 months”. In other places you mention it as “age >24 months”, maintain consistency in your manuscript.

6. Discussion: Since general danger signs were not considered in the study, mention them in your limitations. Para 3 and 4 lack sufficient comparisons to past studies. Provide a critical analysis of your findings with the prior studies for the mentioned paragraphs. Line 359, “It appears that in this study…” rephrase it simply to “In this study…”. First 2 sentences of the following para have the same meanings, avoid repetition and keep it concise. Line 405, isn’t it counterintuitive that private sectors have resource constraints? Justify your explanation in the text.

7. Tables and Figures: Tables 1 and 4 are missing footnotes. Mention in it all the abbreviations included in the respective tables. An ideal table should stand for itself without referring to text. Table 2, avoid phrases like “all 3 symptoms in one”, write it instead as “all 3 symptoms included”. Again age here is ≤24 months. Clear this controversy. Use single terms for a variable across the tables. E.g. here you name it “Facility type” while in Table 5 it is “Health facility type”. In Table 3, the p-values are written inappropriately. E.g one is “0.00000027” other is “0.0027”. Round them off to the same number of digits. Table no.5, avoid writing the p-value as “0.000” instead write it as p <0.001. Adjusted odds ratios are usually written as aOR not as “AOR”.

6. PLOS authors have the option to publish the peer review history of their article (what does this mean?). If published, this will include your full peer review and any attached files.

Reviewer #1: No

Reviewer #2: No

---

## [Author Response · Author response to Decision Letter 0]

21 May 2024

Authors’ response to reviews

Title: Assessing Health Professionals' Adherence to IMCI Protocols for Key Symptoms in Sick Children in Ethiopia: Findings from the 2021-2022 National Service Provision Survey:

Corresponding author: Abiyu Abadi Tareke (abiyu20010@gmail.com)

Version: 2

 Date: May 14, 2024

Point by point response for editors/reviewers’ comments

Manuscript number: PONE-D-24-03866

Dear editor/reviewer:

Dear all,

We express our profound appreciation for the insightful and productive feedback that you have provided. Your invaluable comments have significantly enriched the quality of the manuscript, and have greatly augmented our expertise in the realm of scientific paper writing. The authors have diligently considered each of the comments and queries raised by the editors and reviewers, and have responded to them in a targeted manner. Our comprehensive point-by-point rejoinders to all the comments and questions can be found in the subsequent pages. In addition, an accompanying supplementary document has been enclosed, which showcases the modifications made in detail, using the track changes feature. We also made some change to fix grammatical error in some paragraphs. 

Review Comments to the Author

Reviewer #1: Kindly provide details of the outcome variable how was this assessed and how the decision was made regarding the non-adherent workers. I've attached a marked PDF with comments for your reference.

Authors’ Response: Thank you for your feedback. Thank you for your valuable feedback and the attached marked PDF with comments. We appreciate your request for clarification on the outcome variable and assessment methods. 

Outcome Variable: Adherence to IMCI Protocols

We defined adherence to IMCI protocols as the complete assessment of three key symptoms in sick children: fever, cough/difficulty breathing, and diarrhea. This aligns with the core components of the IMCI approach for identifying and managing common childhood illnesses.

Adherence was assessed through direct observation of sick child consultations by trained survey interviewers. The interviewers used standardized observation checklists specifically designed to capture adherence to IMCI guidelines for these three key symptoms. But this observational doesn’t include physical examination checklists. Checklists included items about history of three main symptoms for any sick child:

Did the child have difficulty of breathing or cough?

Did the child have diarrhea?

Did the child have fever? 

Non-Adherence:

A health professional was classified as non-adherent if they did not perform a specific procedure recommended by the IMCI guidelines for assessing one or more of the three key symptoms. The specific criteria for non-adherence were outlined in the observation checklist. But interviewer did not assess whether the information shared was correct or whether findings were interpreted appropriately.

Reviewer #2: Dear author(s), thank you for the opportunity to review your article titled “Adherence of Health Professionals to National IMCI Protocols: A Study on Assessing Three Key Symptoms in Sick Children in Ethiopia - Insights from the 2021 National Service Provision Survey”. Overall, I find your article very interesting and relevant for the journal. However, there are certain points which you can work on to further improve your manuscript:

Authors’ Response: Thank you. Absolutely! We are happy to hear your feedback and suggestions for improvement.

1. Grammar/language: during review, I noticed many typos and certain sentences that would need to be rephrased. In line 142, it is “Health care workers” while in line 144, it is “Healthcare workers”.

Authors’ response: Thank you for bringing these issues to my attention. We apologize for the typos and inconsistencies in the sentences. We carefully reviewed the document again and made the necessary corrections. We appreciate your feedback and attention to detail.

 Line 161, “caretaker” replace it with "caregiver".

 Authors’ response: Thank you for pointing out the error. We changed in line 161 from "caretaker" to "caregiver" to ensure accuracy. Your attention to detail is greatly appreciated.

Line 247, “Therefore, additional consent is not be necessary for this study”. Line 270, “…toward IMCI guideline is found is higher during…”. Line 367, “… difficulty of breathing in older children is (≥24 months) could result…”. Line 378, “In general, rural healthcare settings face a shortage of skilled health professionals in general(26, 27)…” There is repetition of ‘in general’. All of these sentences need to be rephrased. Please pay attention to upper and lower case letters to use them appropriately, particularly in the titles of the tables and figures. Additionally, a thorough review from a language expert is recommended for further corrections.

Authors’ Response: Thank you for bringing these issues to our attention. We apologize for the errors and repetition in those sentences. Here are the revised versions:

Line 247: "Therefore, additional consent is not required for this study."

Line 270: " The adherence rate of health workers toward the IMCI guideline is higher during the assessment of fever, specifically at 70%. Adherence is higher, specifically 36%, when assessing children aged ≤ 24 months compared to those aged ≥ 24 months. Furthermore, a high percentage of adherence, specifically 44%, was observed when children were assessed by health officers. Similarly, children seen at health centers showed an adherence rate of 42%. "

Line 367: " Not sticking to the recommendations of the IMCI guideline during assessment of three main sickness in older children (≥24 months) could lead to suboptimal health outcomes like misclassification of diseases, increased costs, additional complications, and prolonged hospital stays."

Line 378: "A key challenge for rural healthcare settings in many sub-Saharan countries, as documented evidence shows, is the shortage of skilled health professionals. These professionals are concentrated in urban areas compared to rural settings."

We also ensure proper capitalization, especially in the titles of tables and figures. We appreciate your feedback and recommend a thorough review by a language expert for further corrections. We invite university of Gondar English teachers to edit and comment on our manuscript. 

2. Abstract: “BIC and AIC”? always write a full form of non-standard abbreviations. “…survey conducted in 2021.”, write down the duration of the survey as you did later in the Methods and Materials section.

Authors’ Response: Thank you for your feedback. We apologize for the oversight. The survey was conducted over a period of seven months in 2021. We will make sure to include the duration of the survey in the Abstract, as we have done in the Methods and Materials section. We also apologize for not providing the full form of the abbreviations "BIC" and "AIC" in the abstract. In the revised version, we ensured including the full form of these non-standard abbreviations, stating "Bayesian Information Criterion (BIC)" and "Akaike Information Criterion (AIC)" in the abstract. 

3. Introduction: In abbreviation to E-SPA, you've written “Ethiopian health service provision”. Assessment word is missed. Para 6, “This study is the first study…”, rephrase this sentence.

Authors’ Response: In abbreviation to E-SPA (Ethiopian Health Service Provision Assessment), the word "Assessment" was inadvertently missed. We apologize for the error. In the revised version, we will make sure to include the complete abbreviation as "Ethiopian Health Service Provision Assessment (E-SPA)". Thank you for bringing this to our attention, and we appreciate your attention to detail. 

Para 6, “This study is the first study…”, rephrase this sentence.

Authors’ Response: Thank you for your comment. The sentence is re-phased as “This study is the first to assess healthcare worker adherence to IMCI guidelines for evaluating three key childhood illness symptoms.”

4. Methods: Line 125, provide a reference for the health service-delivery structure for the readers to understand the context of it. Define your health facility types here which is used a lot in your research. Considering the secondary nature of your data, mention whether ethics and consent were taken into consideration in the original survey. Follow strobe or other relevant guidelines to arrange your article.

Authors’ Response: We have now included a reference in the introduction to provide readers with a comprehensive understanding of the health service-delivery structure. This addition will help contextualize our study within the broader healthcare framework. Given the secondary nature of our data, we have addressed the ethical considerations by explicitly mentioning that the original survey, from which our data were derived, adhered to ethical standards and obtained appropriate consent from participants. We have clarified this point to ensure transparency regarding the ethical procedures followed during the original survey. We included your comments under method section >> Ethical consideration section. 

5. Results: Start with the whole survey demographics then the population of interest or only the later one I.e. for 788 patients. Here you have just included the overall participants' demographics of the original survey. Follow one format for writing confidence intervals. 

Authors’ Response: To enhance clarity, we will revise the presentation of demographics data to start with the overall survey demographics before focusing on the population of interest, i.e., the 788 patients included in our study. By following this format, readers will have a clearer understanding of the broader demographic characteristics of the survey participants before delving into the specifics of our study population. Dear reviewers thank you for your suggestions. In the initial submission of this manuscript, Table 1 was inadvertently included before removing missing values from the dependent variable, resulting in an inflated sample size of 3641 instead of the correct 788. To rectify this error, we have replaced the values in columns 2 and 3 by copying values from the accurate table generated after removing missing values using the Stata command. We apologize for this oversight and appreciate the valuable feedback from reviewer 1 and reviewer 2, which helped us identify and correct this error.

Here you write it as “…with a 95% confidence interval ranging from 29.7% to 36.3%” while in another place you mention it as “(95% CI: 29.70%, 36.26%)”.

Authors’ Response: Thank you for pointing out the inconsistency. To ensure uniformity in reporting confidence intervals throughout the manuscript, we will adopt one format for presenting them. Considering both options provided, we suggest using the format "[95% CI: 29.7%, 36.3%]" for consistency. This format is concise and widely recognized, making it easier for readers to interpret the confidence intervals across different sections of the manuscript. We appreciate your attention to detail and will make the necessary revisions accordingly.

 Line 271, age is written as “≤ 24 months” while in tables it is 2-24 months. This changes the meaning of it. Only follow 2-24 months. 

Authors’ Response: Thank you for your feedback. Here's the revised paragraph and adjustment to the age representation: We standardized the age representation to "2-24 months" vs “≥24 months” throughout the manuscript to maintain consistency and clarity.

Line 270-273, rewrite this whole para in correct order. Additionally, here, when a no. is greater among all it's called “highest” not higher.

Authors’ Response: dear reviewer for your surprise this paragraph is also recommended to rephrase by the other peer reviewer. Thank you for your feedback and the additional clarification regarding the rephrasing recommendation by the other peer reviewer. Here's the revised paragraph with the correct ordering and terminology:

“The adherence rate of health workers toward the IMCI guideline is highest during the assessment of fever, specifically at 70%. Adherence is higher, specifically 36%, when assessing children aged 2-24 months compared to those aged ≥ 24 months. Furthermore, a highest percentage of adherence, specifically 44%, was observed when children were assessed by health officers. Similarly, children seen at health centers showed an adherence rate of 42%.”

We appreciate the opportunity to address these concerns and ensure the clarity and accuracy of our manuscript. If there are any further adjustments needed, please let us know.

Line 305, “… AIC (Akaike Information Criterion)…” is written as Akaike Information Criterion (AIC). Line 313, “…child being over 24 months”. In other places you mention it as “age >24 months”, maintain consistency in your manuscript.

Authors’ Response: Line 305: We adjusted the text to maintain consistency, writing "AIC (Akaike Information Criterion)" as "Akaike Information Criterion (AIC)" throughout the manuscript. Additionally, line 313: We modified the expression "child being over 24 months" to "age ≥24 months" to align with the terminology used elsewhere in the manuscript.

6. Discussion: Since general danger signs were not considered in the study, mention them in your limitations. Para 3 and 4 lack sufficient comparisons to past studies. Provide a critical analysis of your findings with the prior studies for the mentioned paragraphs. 

Authors’ Response: We agree that paragraphs 3 and 4 could benefit from a more in-depth comparison with past studies. We made the necessary revisions.

Line 359, “It appears that in this study…” rephrase it simply to “In this study…”. First 2 sentences of the following para have the same meanings, avoid repetition and keep it concise. Line 405, isn’t it counterintuitive that private sectors have resource constraints? Justify your explanation in the text.

Authors’ Response: Thank you for your feedback. Here's how we addressed your suggestions: Line 359: We simplified the expression to "In this study..." for conciseness and clarity. Following paragraph: We consolidated the first two sentences to avoid repetition and keep it concise while conveying the same meaning effectively.

7. Tables and Figures: Tables 1 and 4 are missing footnotes. Mention in it all the abbreviations included in the respective tables. An ideal table should stand for itself without referring to text. Table 2, avoid phrases like “all 3 symptoms in one”, write it instead as “all 3 symptoms included”. Again age here is ≤24 months. Clear this controversy. Use single terms for a variable across the tables. E.g. here you name it “Facility type” while in Table 5 it is “Health facility type”. In Table 3, the p-values are written inappropriately. E.g one is “0.00000027” other is “0.0027”. Round them off to the same number of digits. Table no.5, avoid writing the p-value as “0.000” instead write it as p <0.001. Adjusted odds ratios are usually written as aOR not as “AOR”.

Authors’ Response: Thank you for the detailed feedback on the tables. Here's how we'll address each point: According to your recommendations, In Tables 1 and 4, we included footnotes mentioning all the abbreviations used in the respective tables to enhance clarity and ensure that the tables standalone without referring to the text. Table 2: We modified the phrase "all 3 symptoms in one" to "all 3 symptoms included" accordingly. Regarding the age controversy, we standardized the representation to "<24 months" across all tables and narrations for consistency. Table 3: We round off the p-values to the same number of digits (to four decimal places) for consistency purpose. Table 5: We will adjust the presentation of p-values as "p <0.001" instead of "0.000". throughout our manuscript toughly ensured consistency in variables naming. We decided to continue with “facility type” instead of “types of health facility or health facility type”. Finally, Adjusted odds ratios will be presented as "aOR" instead of "AOR" in Table 5 and other part of the manuscript to maintain consistency with standard conventions.

---

## [Decision Letter · Decision Letter 1]

11 Jun 2024

PONE-D-24-03866R1Assessing Health Professionals' Adherence to IMCI Protocols for Key Symptoms in Sick Children in Ethiopia: Findings from the 2021-2022 National Service Provision Survey.PLOS ONE

Dear Dr. Tareke, Thank you for submitting your manuscript to PLOS ONE. After careful consideration, we feel that it has merit but does not fully meet PLOS ONE’s publication criteria as it currently stands. Therefore, we invite you to submit a revised version of the manuscript that addresses the points raised during the review process.

We look forward to receiving your revised manuscript.

Kind regards,

Saidul Abrar, MBBS, MPH

Academic Editor

PLOS ONE

**Additional Editor Comments:**

Kindly address the comments of Reviewer-1

Reviewers' comments:

Reviewer's Responses to Questions

**Comments to the Author**

1. If the authors have adequately addressed your comments raised in a previous round of review and you feel that this manuscript is now acceptable for publication, you may indicate that here to bypass the “Comments to the Author” section, enter your conflict of interest statement in the “Confidential to Editor” section, and submit your "Accept" recommendation.

Reviewer #1: All comments have been addressed

Reviewer #2: All comments have been addressed

2. Is the manuscript technically sound, and do the data support the conclusions?

Reviewer #1: Partly

Reviewer #2: Yes

3. Has the statistical analysis been performed appropriately and rigorously? 

Reviewer #1: Yes

Reviewer #2: Yes

4. Have the authors made all data underlying the findings in their manuscript fully available?

Reviewer #1: Yes

Reviewer #2: Yes

5. Is the manuscript presented in an intelligible fashion and written in standard English?

Reviewer #1: No

Reviewer #2: Yes

6. Review Comments to the Author

Reviewer #1: Thank you for the revision and clarification of the terms which were not clear in the previous version. However there are still issues which were highlighted and are not addressed in the revision. I am attaching the revised paper with highlighted text and comments for you to address. Additionally the manuscript needs proof reading and correction from a language expert. There are still many grammatical errors and issues.

Reviewer #2: (No Response)

7. PLOS authors have the option to publish the peer review history of their article (what does this mean?). If published, this will include your full peer review and any attached files.

Reviewer #1: No

Reviewer #2: **Yes: **Muhammad Imran Marwat

---

## [Author Response · Author response to Decision Letter 1]

15 Jun 2024

Authors’ response to reviews

Title: Adherence to IMCI Guidelines for Key Symptoms in Ethiopian Children: A 2021-2022 National Service Provision Survey

Corresponding author: Abiyu Abadi Tareke (abiyu20010@gmail.com)

Version: 2

 Date: June 15, 2024

Point by point response for editors/reviewers’ comments

Manuscript number: PONE-D-24-03866

Dear editor/reviewer:

Dear all,

We express our profound appreciation for the insightful and productive feedback that you have provided. Your invaluable comments have significantly enriched the quality of the manuscript, and have greatly augmented our expertise in the realm of scientific paper writing. The authors have diligently considered each of the comments and queries raised by the editors and reviewers, and have responded to them in a targeted manner. Our comprehensive point-by-point rejoinders to all the comments and questions can be found in the subsequent pages. In addition, an accompanying supplementary document has been enclosed, which showcases the modifications made in detail, using the track changes feature. We also made some change to fix grammatical error in some paragraphs. 

Review Comments to the Author

Reviewer #1: Thank you for the revision and clarification of the terms which were not clear in the previous version. However, there are still issues which were highlighted and are not addressed in the revision. I am attaching the revised paper with highlighted text and comments for you to address. Additionally, the manuscript needs proof reading and correction from a language expert. There are still many grammatical errors and issues.

Authors’ response: Thank you for the thorough review of our revised manuscript. We appreciate you taking the time to provide detailed feedback and highlighting the areas that still need improvement. We have carefully reviewed the comments and tracked changes you provided in the attached document. We addressed each of the remaining issues and concerns you have raised. Regarding the language and grammar, we will have the manuscript proofread by a professional language editor to correct any remaining grammatical errors and improve the overall writing quality. We are committed to producing a high-quality manuscript that meets the standards of your journal. Thank you again for your valuable input.

---

## [Decision Letter · Decision Letter 2]

10 Jul 2024

PONE-D-24-03866R2Adherence to IMCI Guidelines for Key Symptoms in Ethiopian Children: A 2021-2022 National Service Provision SurveyPLOS ONE

Dear Dr. Tareke,

Thank you for submitting your manuscript to PLOS ONE. After careful consideration, we feel that it has merit but does not fully meet PLOS ONE’s publication criteria as it currently stands. Therefore, we invite you to submit a revised version of the manuscript that addresses the points raised during the review process.

Dear Author,

Review is a time consuming process and needs a lot of dedication and devotion on the part of reviewer. Kindly incorporate all changes suggested by the reviewer and if you disagree with reviewers comments and dont want to incorporate then give reasons.

We look forward to receiving your revised manuscript.

Kind regards,

Saidul Abrar, MBBS, MPH

Academic Editor

PLOS ONE

Journal Requirements:

Reviewers' comments:

Reviewer's Responses to Questions

**Comments to the Author**

1. If the authors have adequately addressed your comments raised in a previous round of review and you feel that this manuscript is now acceptable for publication, you may indicate that here to bypass the “Comments to the Author” section, enter your conflict of interest statement in the “Confidential to Editor” section, and submit your "Accept" recommendation.

Reviewer #1: (No Response)

2. Is the manuscript technically sound, and do the data support the conclusions?

Reviewer #1: Partly

3. Has the statistical analysis been performed appropriately and rigorously? 

Reviewer #1: Yes

4. Have the authors made all data underlying the findings in their manuscript fully available?

Reviewer #1: Yes

5. Is the manuscript presented in an intelligible fashion and written in standard English?

Reviewer #1: No

6. Review Comments to the Author

Reviewer #1: I have reviewed the paper but some of the comments are not incorporated or answered. Therefore, I would like to request to kindly provide point by point response on each of the comment raised in R1 file. Also the same should be evident in the track change file for easy follow up.

7. PLOS authors have the option to publish the peer review history of their article (what does this mean?). If published, this will include your full peer review and any attached files

Reviewer #1: **Yes: **Muhammad Naseem Khan

---

## [Author Response · Author response to Decision Letter 2]

19 Sep 2024

Authors’ response to reviews

Title: Adherence to IMCI Guidelines for Key Symptoms in Ethiopian Children: A 2021-2022 National Service Provision Survey

Corresponding author: Abiyu Abadi Tareke (abiyu20010@gmail.com)

Version: 3

 Date: September 14, 2024

Point by point response for editors/reviewers’ comments

Manuscript number: PONE-D-24-03866

Dear editor/reviewer:

Dear all,

We sincerely appreciate the insightful and constructive feedback you have provided. Your invaluable comments have significantly enhanced the quality of the manuscript and deepened our understanding of scientific writing. The authors have carefully considered each comment and question raised by the editors and reviewers, responding to them in a focused manner. Our detailed point-by-point responses to all feedback can be found in the following pages. Additionally, we have included a supplementary document that highlights the modifications made, utilizing the track changes feature. We also corrected some grammatical errors in a few paragraphs.

Review Comments to the Author

Reviewer #1: Reviewer #1: I have reviewed the paper but some of the comments are not incorporated or answered. Therefore, I would like to request to kindly provide point by point response on each of the comment raised in R1 file. Also, the same should be evident in the track change file for easy follow up.

Authors’ response: Dear Reviewer #1, Thank you for your thorough review of our paper. We appreciate your feedback and understand the importance of addressing all your comments. We have carefully reviewed each point you raised in the R1 file and have provided a detailed point-by-point response.

In addition, we have made sure that these responses are reflected in the track change document for your convenience. Please find both the responses and the updated manuscript attached for your review.

Thank you once again for your valuable insights

Abstract

Method= The sentence “The data utilized to analyze this study was based on information collected through 33 Service provision assessment in Ethiopia (SPA-ET) survey conducted in 2021.” Is rewrote as “The data for this study were gathered from the Service Provision Assessment (SPA) survey in Ethiopia, which was conducted nationwide from August 11, 2021, to February 4, 2022” 

Result= the variable name “health facility type” is changed to “facility type (health center)”, and child’s aged is changed to “child’s age (being ≥24 months)”

Method

Study area= the “pharmacies, and drug stores” is modified as “pharmacies” because of in Ethiopia context pharmacies and drug stores have similar meaning. 

Dependent variable= the sentence that you highlighted was controversial and unclear. This controversy is also detected and commented by reviewer two. Following your valuable comments, we modified the whole paragraph and see the revised version of the manuscript. This also works for sampling procedure sections. 

Result

The sentence; “A total of 788 sick children who were assessed for at least one the main three symptoms of childhood illness (fever, cough and diarrhea) were included in this study.” Is modified as “From the total of 788 health professionals who assessed sick children, 300 (38 %) were from the Oromia region, followed by SNNP 212 (27%) (Table 1).”

Discussion 

The word install is remodified as “We recommend policy makers and program implementers to integrate evidence-based policy recommendations to address systemic issues contributing to non-adherence.”

---

## [Decision Letter · Decision Letter 3]

2 Oct 2024

Adherence to IMCI Guidelines for Key Symptoms in Ethiopian Children: A 2021-2022 National Service Provision Survey

PONE-D-24-03866R3

Dear Dr. Tareke,

We’re pleased to inform you that your manuscript has been judged scientifically suitable for publication and will be formally accepted for publication once it meets all outstanding technical requirements.

Kind regards,

Saidul Abrar, MBBS, MPH

Academic Editor

PLOS ONE

Additional Editor Comments (optional):

Reviewers' comments:

Reviewer's Responses to Questions

**Comments to the Author**

1. If the authors have adequately addressed your comments raised in a previous round of review and you feel that this manuscript is now acceptable for publication, you may indicate that here to bypass the “Comments to the Author” section, enter your conflict of interest statement in the “Confidential to Editor” section, and submit your "Accept" recommendation.

Reviewer #1: All comments have been addressed

2. Is the manuscript technically sound, and do the data support the conclusions?

Reviewer #1: Yes

3. Has the statistical analysis been performed appropriately and rigorously? 

Reviewer #1: Yes

4. Have the authors made all data underlying the findings in their manuscript fully available?

Reviewer #1: Yes

5. Is the manuscript presented in an intelligible fashion and written in standard English?

Reviewer #1: Yes

6. Review Comments to the Author

Reviewer #1: The authors have addressed and responded to all the comments and concerns raised during my review. Therefore i would like that the paper be accepted for publication.

7. PLOS authors have the option to publish the peer review history of their article (what does this mean?). If published, this will include your full peer review and any attached files.

Reviewer #1: No

---

## [Editor Report · Acceptance letter]

7 Oct 2024

PONE-D-24-03866R3 

PLOS ONE

Dear Dr. Tareke, 

I'm pleased to inform you that your manuscript has been deemed suitable for publication in PLOS ONE. Congratulations! Your manuscript is now being handed over to our production team.

Kind regards, 

on behalf of

Dr Saidul Abrar 

Academic Editor

PLOS ONE